# TRANSFORMERS ARE OPTIMAL EFFECTIVE FIELDS

## ABSTRACT

Are representations in Transformers provably optimal? We present an axiomatic theory of the Transformer architecture. First, we show that a complex-valued Transformer with linear attention and linear feed-forward residual blocks is uniquely determined by a potential field governed by leading free and interactive terms. As practical extensions of the theory, we characterize ReLU/conic/gated MLP and softmax/sparse attention via axiomatic constructions. The implications include a non-exhaustive unification of existing Transformer variants within a single formalism, and a principled foundation for future architecture search.

## 1 INTRODUCTION

Deriving the pivotal Transformer architecture and its many variants (Vaswani et al., 2017; Dehghani et al., 2019; Narang et al., 2021) in a constructive and systematic way naturally motivates the study of its variational form. From CNNs to Transformers, the interactions within the graph of tokens shift from local and deterministic to long-range and dynamic. Applying symmetries to CNNs has inspired geometric deep learning (Bronstein et al., 2021). Likewise, it is necessary to relax geometric constraints on more principled domains whenever they are too restrictive, leaving enough flexibility for the model to learn. This echoes the principle that *good models are those with the least geometric structures.* In the language of physics: *asymmetrical effects must have asymmetrical causes* (Curie, 1894). In machine learning, this idea aligns with *"the bitter lesson"* (Sutton, 2019), which states that models tend to become less hand-designed over time. Another motivation arises from the continual increase in computing power (Moore's Law), which makes it feasible to implement generalist models that do not overfit across a wide variety of test data. Growing computational capacity, and its associated power consumption, encourages algorithm designers to allocate computation within a model as efficiently as possible. Developing a foundational theory of Transformers thus remains an open question of great theoretical and practical value.

**Contribution** We define the notion of *Structural Optimality* in Definition 3 and prove that the self-attention is a natural and minimal interactive term in the effective field immediately after MLP. We then use the theory to predict the necessity of ReLU / conic / gated activation, softmax attention, and sparse attention. The logical chain is:

$$\text{Universal Field} \rightarrow \text{Effective Truncation} \rightarrow \text{Variational Calculus} \rightarrow \text{Optimal Architecture}.$$

## 2 BACKGROUND

**Notation** Let $[n] = \{0, 1, \ldots, n-1\}$ be the set of $n$ indices. The discrete state is written as $x(t, \omega, \sigma) \in \mathbb{F}$, where $t \in \mathcal{T} := [T]$ is the discrete layer index, $\omega \in \Omega := [N]$ indexes tokens (words/pixels), and $\sigma \in \Sigma := [C]$ indexes neurons or vector field components. $\mathbb{F}$ is a number field either real $\mathbb{R}$ or complex $\mathbb{C}$. The symbols $T$, $N$, and $C$ denote the depth, the sequence length (number of tokens), and the feature space dimension (network width), respectively. In a continuous-layer setting we inherit the notation $\mathcal{T} := [0, 1]$, and write $\dot{x} := \frac{\mathrm{d}x}{\mathrm{d}t}$ for the layer-wise derivative and omit the layer as an implicit variable. For brevity, we assume $\Sigma, \Omega$ are finite and denote the discrete state as a layer-dependent matrix $\boldsymbol{X} \in \mathbb{F}^{N \times C}$ whose row and column entries are $\boldsymbol{X}_{\omega\sigma} := x(\omega, \sigma)$, and denote each token at $\omega \in \Omega$ as a vector $x_\omega \in \mathbb{F}^C$ in the feature space whose components are $(x_\omega)_\sigma := x(\omega, \sigma)$. The parameters are the layer-dependent matrices $\boldsymbol{W} = \boldsymbol{W}(t) \in \mathbb{F}^{C \times C}$. We use $\odot$ for element-wise (Hadamard) operations. The Lie bracket $[f, g] := fg - gf$ denotes the commutator. $\boldsymbol{W}^\top$ and $\boldsymbol{W}^*$ are the real and conjugate / Hermitian transposes of $\boldsymbol{W}$ respectively. $\|\cdot\|_\mathrm{F}$ denotes the

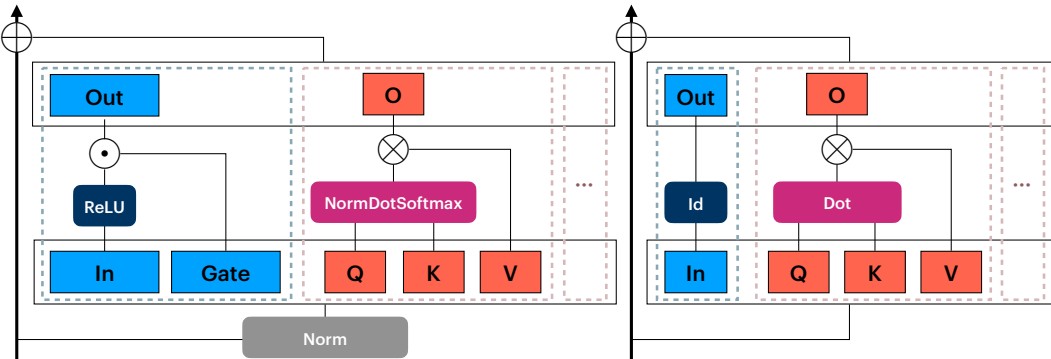

Figure 1: **Left:** a practical parallelized Transformer block with GLU activation. The wide MLP can be equivalently separated into multiple narrow heads. **Right:** a simplified Transformer block eliminating activation functions, normalizations and softmax.

Frobenius norm. $U(C)$ is the $C$-dimensional unitary group, $S_N$ is the $N$-dimensional permutation group, and $C_{U(C)}(\boldsymbol{W}) := \{\boldsymbol{V} \in U(C) : [\boldsymbol{V}, \boldsymbol{W}] = 0\}$ is the centralizer of $\boldsymbol{W}$ within $U(C)$.

## 2.1 Transformer as ODE

Figure 1 (left) visualizes the architecture of a parallelized Transformer (Dehghani et al.) as a two-layer ResNet with mixed nonlinearities. Suppose the parameters of the $i$-th attention head are $\boldsymbol{Q}_i, \boldsymbol{K}_i, \boldsymbol{V}_i, \boldsymbol{O}_i \in \mathbb{F}^{C \times C_A}$, the parameters of the MLP are $\boldsymbol{W}_{\text{gate}}, \boldsymbol{W}_{\text{in}}, \boldsymbol{W}_{\text{out}} \in \mathbb{F}^{C \times C_{\text{MLP}}}$, and the number field is real $\mathbb{F} = \mathbb{R}$. Layer dependency is implicit in the notation, and normalization is ommited as it is approximately linear in high dimension.

**Definition 1** (Transformers). The ODE form of a Transformer is defined as

$$\dot{\boldsymbol{X}} = ((\boldsymbol{X}\boldsymbol{W}_{\text{gate}}) \odot \text{ReLU}(\boldsymbol{X}\boldsymbol{W}_{\text{in}}))\boldsymbol{W}_{\text{out}} + \sum_{i=1}^{n} \text{Softmax}(C_A^{-1/2}\boldsymbol{X}\boldsymbol{Q}_i\boldsymbol{K}_i^\top\boldsymbol{X}^\top)\boldsymbol{X}\boldsymbol{V}_i\boldsymbol{O}_i^\top. \quad (1)$$

It can be more concisely written as $\dot{\boldsymbol{X}} = \lambda(\boldsymbol{X}\boldsymbol{W}_{\text{all}})\boldsymbol{W}'_{\text{all}}$ where $\lambda := \lambda_{\text{MLP}} \oplus \lambda_A$, or in expanded form, $\dot{\boldsymbol{X}} = \lambda_{\text{MLP}}(\boldsymbol{X}\boldsymbol{W}_{\text{MLP}})\boldsymbol{W}'_{\text{MLP}} + \lambda_A(\boldsymbol{X}\boldsymbol{W}_A)\boldsymbol{W}'_A$, where $\boldsymbol{W}_{\text{all}} := [\boldsymbol{W}_{\text{MLP}} \mid \boldsymbol{W}_A] = [\boldsymbol{W}_{\text{gate}} \mid \boldsymbol{W}_{\text{in}} \mid \boldsymbol{Q}_1 \mid \boldsymbol{K}_1 \mid \boldsymbol{V}_1 \mid \ldots \mid \boldsymbol{Q}_n \mid \boldsymbol{K}_n \mid \boldsymbol{V}_n]$ and $\boldsymbol{W}'_{\text{all}} := [\boldsymbol{W}'_{\text{MLP}} \mid \boldsymbol{W}'_A] = [\boldsymbol{W}_{\text{out}} \mid \boldsymbol{O}_1 \mid \ldots \mid \boldsymbol{O}_n]$.

The more common Transformer architecture is a discrete-layer operator-splitting scheme of this ODE. The split scheme is approximately equal to the parallelized scheme when the nonlinearities commute. Otherwise they are approximately equal when the model is deep.

## 2.2 Comparison between Parallelized and Split Transformers

The discrete layer is a forward (Euler) step $\boldsymbol{X}(t+1) = \boldsymbol{X}(t) + \lambda(\boldsymbol{X}(t)\boldsymbol{W}(t))\boldsymbol{W}'(t)$ and the following approximation links between parallelized and common Transformers.

$$(1 + h\lambda_{\text{MLP}} + h\lambda_A) \approx e^{h\lambda_{\text{MLP}} + h\lambda_A} \approx e^{h\lambda_{\text{MLP}}}e^{h\lambda_A} \approx (1 + h\lambda_{\text{MLP}})(1 + h\lambda_A) \quad (2)$$

In Equation 2, the left/right hand side is a layer of the parallelized/split Transformer respectively. The first and last approximations become equal as $h \to 0$ whereas the second equality holds if and only if $\lambda_{\text{MLP}}$ and $\lambda_A$ commute ($[\lambda_{\text{MLP}}, \lambda_A] = 0$). In the general case when the ODE generators don't commute, deeper models are beneficial in mixing the nonlinear terms, summarized as Lemma 2, as a consequence of Lemma 1.

**Lemma 1** (Lie-Trotter). *The equality holds:* $e^{\lambda_{MLP} + \lambda_A} = \lim_{T \to \infty}(e^{\frac{\lambda_{MLP}}{T}} e^{\frac{\lambda_A}{T}})^T$.

**Lemma 2** (The Benefit of Depth). *Suppose* $\lambda_{MLP}^{\boldsymbol{W}(t)}(\boldsymbol{X}) := \lambda_{MLP}(\boldsymbol{X}\boldsymbol{W}(t))\boldsymbol{W}'(t)$ *and* $\lambda_A^{\boldsymbol{W}(t)}(\boldsymbol{X}) := \lambda_A(\boldsymbol{X}\boldsymbol{W}(t))\boldsymbol{W}'(t)$, *then the parallel and the split schemes asymptotically coincide:*

$$\lim_{T \to \infty} \prod_{t=0}^{T-1}(1 + \lambda_{MLP}^{\boldsymbol{W}(t/T)})(1 + \lambda_A^{\boldsymbol{W}(t/T)}) = \lim_{T \to \infty} \prod_{t=0}^{T-1}(1 + \lambda_{MLP}^{\boldsymbol{W}(t/T)} + \lambda_A^{\boldsymbol{W}(t/T)}).$$

The non-commutative correction can be used to formulate sparse attention in Appendix A.

## 2.3 VARIATIONAL FORMULATION

To formulate the optimality of the ODE, we recall its potential/action/variational/path-integral form.

**Definition 2** (Potential/Action). The (potential) field, whenever it exists, of an ODE path $\boldsymbol{X}$ : $[0,1] \to \mathbb{C}^{N \times C}$ is a function $V : \mathbb{C}^{N \times C} \to \mathbb{C}$ such that the ODE coincides with the Euler-Lagrange equation $\dot{\boldsymbol{X}} = \frac{\partial V}{\partial \boldsymbol{X}^*}$ (or equivalently $-\dot{\boldsymbol{X}}^* = \frac{\partial V}{\partial \boldsymbol{X}}$) that extremizes the action of the path defined as

$$S(\boldsymbol{X}) := \int_0^1 \mathrm{Tr}(\frac{1}{2}(\dot{\boldsymbol{X}}\boldsymbol{X}^* - \boldsymbol{X}\dot{\boldsymbol{X}}^*) - V(\boldsymbol{X})) \, \mathrm{d}t. \tag{3}$$

The real and imaginary parts of $V$ are the dissipative and conservative parts respectively. Note that $\alpha := \frac{1}{2}(\mathrm{d}\boldsymbol{X}\boldsymbol{X}^* - \boldsymbol{X}\,\mathrm{d}\boldsymbol{X}^*)$ is canonical, since $\mathrm{d}\alpha = \mathrm{d}\boldsymbol{X} \wedge \boldsymbol{X}^*$ is the symplectic form, and if we set $\boldsymbol{X} = \frac{q+\mathrm{i}p}{\sqrt{2}}$, then $\max_{\dot{q}} \frac{1}{2}(\dot{\boldsymbol{X}}\boldsymbol{X}^* - \boldsymbol{X}\dot{\boldsymbol{X}}^*) - V(\dot{q}) = p\dot{q} - V(\dot{q})$ recovers the Legendre transform.

**Definition 3** (Structural Optimality). An architecture $\boldsymbol{X}$ whose input and output are $\boldsymbol{X}(0)$ and $\boldsymbol{X}(1)$ is said to be *structurally optimal* with respect to a field $V$ if it extremizes the action of the path $\boldsymbol{X}(t)$.

## 3 OPTIMALITY OF TRANSFORMERS

We first study a complex ODE whose potential exists, visualized in Figure 1 (right) before discussing engineering-meaningful cases to simplify the theory.

### 3.1 LINEARIZED TRANSFORMERS

Consider the complex-valued case $\mathbb{F} = \mathbb{C}$, and use the Hermitian transpose instead of the transpose. Then consider a simplified Transformer without nonlinear functions (activation functions, softmax, etc.) and without gating layers.

**Definition 4** (Linearized Transformers). A linearized Transformer ODE is defined as the equation

$$\dot{\boldsymbol{X}} = \boldsymbol{X}\boldsymbol{W}_{\mathrm{in}}\boldsymbol{W}_{\mathrm{out}}^* + \sum_{i=1}^n C_{\mathrm{A}}^{-1/2} \boldsymbol{X}\boldsymbol{Q}_i\boldsymbol{K}_i^*\boldsymbol{X}^*\boldsymbol{X}\boldsymbol{V}_i\boldsymbol{O}_i^*. \tag{4}$$

From this definition, we can convert the ODE into a potential field form by the following lemma.

**Lemma 3** (Linearized Transformers As Fields). *Suppose $n$ is even, and then there exists parameters $\boldsymbol{W}_{in}, \boldsymbol{W}_{out}, \boldsymbol{Q}_i, \boldsymbol{K}_i, \boldsymbol{V}_i$ such that Equation 4 is associated with the field*

$$V(\boldsymbol{X}) = \mathrm{Tr}(\boldsymbol{X}\boldsymbol{W}_{MLP}\boldsymbol{X}^*) + \frac{1}{\sqrt{C_A}} \sum_{i=1}^{n/2} \mathrm{Tr}(\boldsymbol{X}\boldsymbol{W}_{Ai}\boldsymbol{X}^*\boldsymbol{X}\boldsymbol{W}_{Bi}^*\boldsymbol{X}^*).$$

Note that $\boldsymbol{Q}_i\boldsymbol{K}_i^* = \boldsymbol{V}_{i+n/2}\boldsymbol{O}_{i+n/2}^*$ and $\boldsymbol{Q}_{i+n/2}\boldsymbol{K}_{i+n/2}^* = \boldsymbol{V}_i\boldsymbol{O}_i^*$, otherwise the potential is not guaranteed to exist; we leave the general case for future work. Likewise, we may also recover the softmax function out of the free energy potential in Appendix A.

### 3.2 OPTIMALITY OF MATRIX POTENTIALS

To formalize the optimality of the potential, we assume necessary axioms:

**Axiom 1** (Universality). $V(\boldsymbol{X})$ is analytic in $(\mathrm{Re}\boldsymbol{X}, \mathrm{Im}\boldsymbol{X})$.

**Axiom 2** (Isometry Invariance). At each $t$, $\forall \boldsymbol{P} \in U(N)$, there holds $V(\boldsymbol{P}\boldsymbol{X}) = V(\boldsymbol{X})$.

**Axiom 3** (Learnable Hamiltonians). At each $t$, there exist a set of learnable parameter matrices $\boldsymbol{W}_1, \ldots, \boldsymbol{W}_n \in \mathbb{C}^{C \times C}$: $\forall \boldsymbol{R} \in \bigcap_{i=1}^n C_{U(C)}(\boldsymbol{W}_i)$, there holds $V(\boldsymbol{X}\boldsymbol{R}^*) = V(\boldsymbol{X})$.

**Axiom 4** (Optional, Bottleneck). $C_{\mathrm{A}} := \max_{1 \le i \le n} \mathrm{Rank}(\boldsymbol{W}_i) \ll C$.

**Intuitions** Axiom 1 reflects the standard assumption in effective field theory that interactions are Taylor series; higher order terms approximately vanish by renormalization scaling. Axiom 2 ensures that the energy depends only on the intrinsic geometry (Gram matrix $\boldsymbol{X}^*\boldsymbol{X}$), which generalizes token exchangeability to basis independence, since $S_N \subset U(N)$. Axiom 3 asserts that the feature space is isotropic (symmetric) except for the directions distinguished by the interaction matrices ($\boldsymbol{R}^*\boldsymbol{W}\boldsymbol{R} = \boldsymbol{W}$ for all $\boldsymbol{R} \in U(C)$). Axiom 4 is optional, reflecting the symmetry-breaking principle (Curie, 1894), by assuming that interactions are mediated through low-dimensional channels.

**Lemma 4** (Canonical Form of Matrix Fields). *Under Axioms 1, 2, and 3, the matrix potential takes the form of $V(\boldsymbol{X}) = \operatorname{Tr} f(\boldsymbol{X}\boldsymbol{W}_1\boldsymbol{X}^*, \ldots, \boldsymbol{X}\boldsymbol{W}_n\boldsymbol{X}^*)$ for some analytic spectral function $f : \mathbb{C}^n \to \mathbb{C}$ and some parameter matrices $\boldsymbol{W}_1, \ldots, \boldsymbol{W}_n \in \mathbb{C}^{C \times C}$.*

*Sketch of Proof.* First we check that the axioms hold for the form of $V$: for all $\boldsymbol{P}, \boldsymbol{R}$ in the condition, by definitions of unitary group and centralizers, we have $V(\boldsymbol{P}\boldsymbol{X}\boldsymbol{R}^*) = \operatorname{Tr} f(\{\boldsymbol{P}\boldsymbol{X}\boldsymbol{R}^*\boldsymbol{W}_i\boldsymbol{R}\boldsymbol{X}^*\boldsymbol{P}^*\}_{i=1}^n) = \operatorname{Tr} f(\{\boldsymbol{X}\boldsymbol{W}_i\boldsymbol{X}^*\}_{i=1}^n) = V(\boldsymbol{X})$. Next we show the uniqueness of the form of $V$. By Axiom 1, the matrix potential can be written as an infinite polynomial series $V : \mathbb{C}^{N \times C} \to \mathbb{C}$. By Axiom 2, the potential $V(\boldsymbol{X}) = V(\boldsymbol{X}^*\boldsymbol{X}) : \mathbb{C}^{C \times C} \to \mathbb{C}$. By adding Axiom 3, the potential $V(\boldsymbol{X}) = \operatorname{Tr} f(\boldsymbol{X}\boldsymbol{W}_1\boldsymbol{X}^*, \boldsymbol{X}\boldsymbol{W}_2\boldsymbol{X}^*, \ldots, \boldsymbol{X}\boldsymbol{W}_n\boldsymbol{X}^*)$ for $f : \mathbb{C}^n \to \mathbb{C}$. (Details in Appendix B.) □

**Theorem 5** (Optimality of Linearized Transformers). *Under Axioms 1, 2, 3, Definition 4 is the minimal nontrivial interaction ODE.*

The implication of Theorem 5 is that the linearized Transformer in Equation 4 is a minimal model, i.e. it is the *effective* interactive field. Softmax attention in Equation 1 from the free energy potential and many variants (powers, sigmoid, etc.) also satisfies the canonical form in Lemma 4 with higher-order terms. Finally, we comment that the width of the MLP and attention matrices are a result of the low-rank assumption.

**Lemma 6** (Low-Rankness). *Under Axiom 4, if $n$ is sufficiently large, then $C_A \ll C < C_{MLP}$.*

## 4 RELATED WORKS

Martin & Hinrichs (2025) studied deep models' generalization by layer-wise spectral overlap between a varying $\boldsymbol{W}_S(t)$ and a fixed $\boldsymbol{W}_T(t)$, which naturally compares to our complex Hilbert space product. Barnfield et al. (2025) extended the double-descent curve of test error over aspect ratio from the MLP to the self-attention case. Furuya et al. (2025) proved a depth-dependent universality bound in terms of approximation theory, whereas our work discusses the effective approximation within a "universal" function class. Marcotte et al. (2025) studied the parameter symmetry induced by the activation function and self-attention, which connects to our symmetry axioms.

## 5 CONCLUSION AND PERSPECTIVES

We have shown a constructive proof of the Transformer architecture from variational principles and theoretical/quantum mechanics, which offers a guide to study the residual stream geometry for the mechanistic interpretability of in-context learning (such as harmonic (Kantamneni & Tegmark, 2025) or rapid oscillation (Patrawala et al., 2025)). More structures including mixture-of-expert MLPs and non-attentional models (MLP-Mixers, State Space Models, etc) are left for future work. The applications of the paper are listed in Appendix A.

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

## A   ENGINEERING IMPLICATIONS

In this section, we exemplify the usefulness of the theoretical framework when it extends to engineering-meaningful architectures by adapting to different symmetries.

**Activation-MLP as Projected Gradient**   In the MLP $\lambda^{\boldsymbol{W}(t)}(\boldsymbol{X}) = \sum_{i=1}^{n}(\boldsymbol{X}\boldsymbol{W}_{\text{in},i})_{+}\boldsymbol{W}_{\text{out},i}^{*}$, ReLU is a projection towards the positive cones (orthants) $\mathbb{R}_{+}^{C_{\text{A}}}$. In the space of the ODE path $\boldsymbol{X}$, the cones are affinely transformed to $\boldsymbol{W}_{\text{in},i}\mathbb{R}_{+}^{C_{\text{A}}}\boldsymbol{W}_{\text{out},i}^{*}$, whose directions and sections are learnable. In the convex optimization language, $\text{ReLU}(\boldsymbol{X}) = \Pi_{\mathbb{R}_{+}}(\boldsymbol{X}) := \min_{\boldsymbol{Y}\in\mathbb{R}_{+}}\|\boldsymbol{X}-\boldsymbol{Y}\|_{\text{F}}^{2}$ is the projection operator and $\Pi_{\boldsymbol{W}_{\text{MLP}}}(\boldsymbol{X}) := (\boldsymbol{X}\boldsymbol{W}_{\text{in}})_{+}\boldsymbol{W}_{\text{out}}^{*}$ is the affine projection onto the affine orthant parameterized by $\boldsymbol{W}_{\text{in}}$ and $\boldsymbol{W}_{\text{out}}$.

As one of the predictions of our theory, previous work (Fu & Cohen, 2024) showed that ReLU is geometrically characterized by three axioms: **idempotence** ($\lambda \circ \lambda = \lambda$), **positive homogeneity** ($\lambda \circ h = h \circ \lambda, \forall h > 0$), and **permutation equivariance** ($\pi \circ \lambda = \lambda \circ \pi, \forall \pi \in S_{C}$). The third axiom restricts the form of $\lambda$ to be component-wise. Relaxing the symmetry group $S_{C}$ to subspace rotation group $S_{n} \times U(C_{\text{A}})$, where $n = C/C_{\text{A}}$ is the number of cones / heads, improves general-purpose model generalization and learning efficiency including GPTs, ResNets and Diffusion Transformers.

**Gated MLP as Minimal Permutation-Invariant Field**   As we explained in the intuition of Axiom 3, the left unitary equivariance is a stronger condition than token exchangeability. Once we relax $U(N)$ to $S_{N}$, the form of $V$ is no longer restricted to $V(\boldsymbol{X}^{*}\boldsymbol{X})$, but a row-wise function is sufficient. The minimal *nonlinear* model in this case is the gated MLP without ReLU $\dot{\boldsymbol{X}} = (\boldsymbol{X}\boldsymbol{W}_{\text{gate}}) \odot (\boldsymbol{X}\boldsymbol{W}_{\text{in}})\boldsymbol{W}_{\text{out}}^{\top}$ recovered from the triple-trace potential $V(\boldsymbol{X}) = \frac{1}{3}\sum_{jk}((\boldsymbol{X}\boldsymbol{W}_{\text{gate}}) \odot (\boldsymbol{X}\boldsymbol{W}_{\text{in}}) \odot (\boldsymbol{X}\boldsymbol{W}_{\text{out}}))_{jk}$ and $\dot{\boldsymbol{X}} = \partial V/\partial \boldsymbol{X}$. We may also recover GLU (Shazeer, 2020) by the ReLU projected gradient flow, and the $\text{ReLU}^{2}$ activation by projecting and tying the input and gating parameters.

**Softmax Attention as Free-Energy Field**

**Lemma 7** (Softmax Transformers). *Suppose $n$ is even. There exist parameters such that the ODE $\dot{\boldsymbol{X}} = \sum_{i=1}^{n} \text{Softmax}(C_{A}^{-1/2}\boldsymbol{X}\boldsymbol{Q}_{i}\boldsymbol{K}_{i}^{*}\boldsymbol{X}^{*})\boldsymbol{X}\boldsymbol{V}_{i}\boldsymbol{O}_{i}^{*}$ is associated with the free energy ("log-sum-exp") potential $V(\boldsymbol{X}) = \sqrt{C_{A}}\sum_{i=1}^{n}\sum_{j=1}^{N}\log\sum_{k=1}^{N}\exp([\frac{1}{\sqrt{C_{A}}}\boldsymbol{X}\boldsymbol{W}_{Ai}\boldsymbol{X}^{*}]_{jk})$.*

Note that the potential exists only when $\boldsymbol{Q}_{i}\boldsymbol{K}_{i}^{*} = \boldsymbol{V}_{i}\boldsymbol{O}_{i}^{*} = (\boldsymbol{Q}_{i+n/2}\boldsymbol{K}_{i+n/2}^{*})^{*} = (\boldsymbol{V}_{i+n/2}\boldsymbol{O}_{i+n/2}^{*})^{*}$, and we leave the general case as a future work.

**Sparse Attention as Non-Commutative Correction**

**Lemma 8** (Sparse Attention). *Under the pairwise non-commutative condition $[\lambda_{MLP}, \lambda_{A}] = \beta\lambda_{A}$, the sequential composition of the layers is equivalent to a single effective flow generated by:*

$$\exp(\lambda_{MLP})\exp(\lambda_{A}) = \exp(\lambda_{MLP} + G(\beta) \cdot \lambda_{A})$$

*where the effective interaction gate $G(\beta) = \frac{\beta}{1-e^{-\beta}}$.*

**Positional Encodings Do Not Violate Symmetry**   Rotary positional encodings (Heo et al., 2024) are Fourier bases $\boldsymbol{X}_{\omega\sigma} = e^{i\omega\exp(-T_{\text{PE}}\sigma/C)} \in U(1) \subset \mathbb{C}$ where the temperature can be $T_{\text{PE}} = \log(10^{9})$ in modern implementations. Tokens with positional encoding dimensions do not break permutation symmetry among tokens, but since they depend on token index $\omega$, spatial pairwise / outer products reflect relative distances between words.

## B   PROOFS

**Lemma 1** (Lie-Trotter). *The equality holds: $e^{\lambda_{MLP}+\lambda_{A}} = \lim_{T\to\infty}(e^{\frac{\lambda_{MLP}}{T}}e^{\frac{\lambda_{A}}{T}})^{T}$.*

*Proof.* Applying the Baker-Campbell-Hausdorff expansion (Varadarajan, 2013) $e^{\lambda_{\text{MLP}}t}e^{\lambda_{\text{A}}t} = e^{(\lambda_{\text{MLP}}+\lambda_{\text{A}})t+\frac{1}{2}[\lambda_{\text{MLP}},\lambda_{\text{A}}]t^{2}+O(t^{3})}$ gives the leading non-commutative correction. □

**Lemma 2** (The Benefit of Depth). *Suppose* $\lambda_{MLP}^{\boldsymbol{W}(t)}(\boldsymbol{X}) := \lambda_{MLP}(\boldsymbol{XW}(t))\boldsymbol{W}'(t)$ *and* $\lambda_A^{\boldsymbol{W}(t)}(\boldsymbol{X}) :=$
$\lambda_A(\boldsymbol{XW}(t))\boldsymbol{W}'(t)$, *then the parallel and the split schemes asymptotically coincide:*

$$\lim_{T\to\infty}\prod_{t=0}^{T-1}(1+\lambda_{MLP}^{\boldsymbol{W}(t/T)})(1+\lambda_A^{\boldsymbol{W}(t/T)}) = \lim_{T\to\infty}\prod_{t=0}^{T-1}(1+\lambda_{MLP}^{\boldsymbol{W}(t/T)}+\lambda_A^{\boldsymbol{W}(t/T)}).$$

*Proof.* This result follows from Lemma 1 and the convergence properties of discretization schemes for the ODE $\dot{\boldsymbol{X}} = \lambda_{\text{MLP}}^{\boldsymbol{W}(t)}(\boldsymbol{X}) + \lambda_{\text{A}}^{\boldsymbol{W}(t)}(\boldsymbol{X})$, where the parameters $\boldsymbol{W}(t)$ make the operators time-dependent.

The parallel scheme (right-hand side) corresponds to the standard Euler discretization with step size $1/T$. Setting $t_k = k/T$, each term is $1 + \frac{1}{T}(\lambda_{\text{MLP}}^{\boldsymbol{W}(t_k)} + \lambda_{\text{A}}^{\boldsymbol{W}(t_k)}) + O(1/T^2)$. The product over $T$ steps approximates the time-ordered exponential $\mathcal{T}\exp\left(\int_0^1(\lambda_{\text{MLP}}^{\boldsymbol{W}(t)}(\boldsymbol{X}) + \lambda_{\text{A}}^{\boldsymbol{W}(t)}(\boldsymbol{X}))\,dt\right)$ as $T \to \infty$, with a global error of $O(1/T)$.

The split scheme (left-hand side) is a split-step Euler discretization. Expanding the product for a single step yields:

$$\left(1+\frac{1}{T}\lambda_{\text{MLP}}^{\boldsymbol{W}(t_k)}\right)\left(1+\frac{1}{T}\lambda_{\text{A}}^{\boldsymbol{W}(t_k)}\right) = 1+\frac{1}{T}\left(\lambda_{\text{MLP}}^{\boldsymbol{W}(t_k)}+\lambda_{\text{A}}^{\boldsymbol{W}(t_k)}\right)+\frac{1}{T^2}\left(\lambda_{\text{MLP}}^{\boldsymbol{W}(t_k)}\lambda_{\text{A}}^{\boldsymbol{W}(t_k)}\right).$$

This matches the parallel scheme with an extra interaction term $\frac{1}{T^2}(\lambda_{\text{MLP}}^{\boldsymbol{W}(t_k)}\lambda_{\text{A}}^{\boldsymbol{W}(t_k)})$. Since this local error is $O(1/T^2)$, the cumulative error over $T$ steps is $T \times O(1/T^2) = O(1/T)$, which vanishes as $T \to \infty$. Thus, by the Lie-Trotter formula (Lemma 1), both schemes converge to the same limit. $\square$

**Lemma 3** (Linearized Transformers As Fields). *Suppose $n$ is even, and then there exists parameters* $\boldsymbol{W}_{in}, \boldsymbol{W}_{out}, \boldsymbol{Q}_i, \boldsymbol{K}_i, \boldsymbol{V}_i$ *such that Equation 4 is associated with the field*

$$V(\boldsymbol{X}) = \text{Tr}(\boldsymbol{XWX}^*) + \frac{1}{\sqrt{C_A}}\sum_{i=1}^{n/2}\text{Tr}(\boldsymbol{XW}_{Ai}\boldsymbol{X}^*\boldsymbol{XW}_{Bi}^*\boldsymbol{X}^*).$$

*Proof.* We separate the potential into linear and interaction terms $V = V_{\text{lin}} + V_{\text{int}}$, and compute the Wirtinger derivative $\partial V/\partial \boldsymbol{X}^*$ to show it matches $\dot{\boldsymbol{X}}$. Treat $\boldsymbol{X}$ and $\boldsymbol{X}^*$ as independent variables.

For the linear term, the differential $\text{d}V_{\text{lin}} = \text{Tr}(\text{d}\boldsymbol{XWX}^*) + \text{Tr}(\boldsymbol{XW}\,\text{d}\boldsymbol{X}^*)$. The term with $\text{d}\boldsymbol{X}^*$ is $\text{Tr}(\boldsymbol{XW}\,\text{d}\boldsymbol{X}^*) = \text{Tr}(\text{d}\boldsymbol{X}^*\boldsymbol{XW})$, so $\partial V_{\text{lin}}/\partial \boldsymbol{X}^* = \boldsymbol{XW}$.

For the interaction term, consider one term $\text{Tr}(\boldsymbol{XAX}^*\boldsymbol{XBX}^*)$. The differential is

$$\text{Tr}(\text{d}\boldsymbol{XAX}^*\boldsymbol{XBX}^*)+\text{Tr}(\boldsymbol{XA}\,\text{d}\boldsymbol{X}^*\boldsymbol{XBX}^*)+\text{Tr}(\boldsymbol{XAX}^*\,\text{d}\boldsymbol{XBX}^*)+\text{Tr}(\boldsymbol{XAX}^*\boldsymbol{XB}\,\text{d}\boldsymbol{X}^*).$$

The terms with $\text{d}\boldsymbol{X}^*$ are

$$\text{Tr}(\boldsymbol{XA}\,\text{d}\boldsymbol{X}^*\boldsymbol{XBX}^*) = \text{Tr}(\text{d}\boldsymbol{X}^*\boldsymbol{XBX}^*\boldsymbol{XA})$$
$$\text{Tr}(\boldsymbol{XAX}^*\boldsymbol{XB}\,\text{d}\boldsymbol{X}^*) = \text{Tr}(\text{d}\boldsymbol{X}^*\boldsymbol{XAX}^*\boldsymbol{XB}).$$

Thus, the contribution to $\partial V_{\text{int}}/\partial \boldsymbol{X}^*$ is $\frac{1}{\sqrt{C_A}}\sum_{i=1}^{n/2}(\boldsymbol{XW}_{\text{B}i}^*\boldsymbol{X}^*\boldsymbol{XW}_{\text{A}i} + \boldsymbol{XW}_{\text{A}i}\boldsymbol{X}^*\boldsymbol{XW}_{\text{B}i}^*)$. Overall, $\partial V/\partial \boldsymbol{X}^* = \boldsymbol{XW} + \frac{1}{\sqrt{C_A}}\sum_{i=1}^{n/2}(\boldsymbol{XW}_{\text{B}i}^*\boldsymbol{X}^*\boldsymbol{XW}_{\text{A}i} + \boldsymbol{XW}_{\text{A}i}\boldsymbol{X}^*\boldsymbol{XW}_{\text{B}i}^*)$. Now take $\boldsymbol{W} = \boldsymbol{W}_{\text{in}}\boldsymbol{W}_{\text{out}}^*$, $\boldsymbol{W}_{\text{A}i} = \boldsymbol{Q}_i\boldsymbol{K}_i^*$, $\boldsymbol{W}_{\text{B}i} = \boldsymbol{V}_i\boldsymbol{O}_i^*$. With the pairing condition $\boldsymbol{Q}_i\boldsymbol{K}_i^* = \boldsymbol{V}_{i+n/2}\boldsymbol{O}_{i+n/2}^*$ and $\boldsymbol{Q}_{i+n/2}\boldsymbol{K}_{i+n/2}^* = \boldsymbol{V}_i\boldsymbol{O}_i^*$ (ensuring symmetry), the derivative matches $\dot{\boldsymbol{X}}$ for the ODE. $\square$

**Lemma 4** (Canonical Form of Matrix Fields). *Under Axioms 1, 2, and 3, the matrix potential takes the form of* $V(\boldsymbol{X}) = \text{Tr}f(\boldsymbol{XW}_1\boldsymbol{X}^*,\ldots,\boldsymbol{XW}_n\boldsymbol{X}^*)$ *for some analytic spectral function* $f : \mathbb{C}^n \to \mathbb{C}$.

*Proof.* First, we check that the axioms hold for the form of $V$: for all $\boldsymbol{P}, \boldsymbol{R}$ in the condition, by definitions of unitary group and centralizers, we have

$$V(\boldsymbol{PXR}^*) = \text{Tr}f(\{\boldsymbol{PXR}^*\boldsymbol{W}_i\boldsymbol{RX}^*\boldsymbol{P}^*\}_{i=1}^n)$$
$$= \text{Tr}f(\{\boldsymbol{XW}_i\boldsymbol{X}^*\}_{i=1}^n)$$
$$= V(\boldsymbol{X}).$$

Next, we show the uniqueness of the form of $V$. Step 1: by Axiom 1, the matrix potential has an expansion form $V(\boldsymbol{X}) = f_1(\boldsymbol{X})$ for some analytic $f_1 : \mathbb{C}^{N \times C} \to \mathbb{C}$.

Step 2: Suppose the polar decomposition of $\boldsymbol{X}$ is $\boldsymbol{X} = \boldsymbol{U}\boldsymbol{D}$ where $\boldsymbol{U} \in U(N)$ and $\boldsymbol{D} = \sqrt{\boldsymbol{X}^*\boldsymbol{X}}$ is positive semi-definite. With Axiom 2, taking $\boldsymbol{P} = U^* \in U(N)$, we have $V(\boldsymbol{U}\boldsymbol{X}) = V(\boldsymbol{U}^*\boldsymbol{X}) = V(\boldsymbol{U}^*\boldsymbol{U}\boldsymbol{D}) = V(\boldsymbol{D})$, which depends solely on $\boldsymbol{D}^2 := X^*X$.

Step 3: We seek quantities $I(\boldsymbol{X})$ such that $I(\boldsymbol{X}\boldsymbol{R}^*) = I(\boldsymbol{X})$ for all $\boldsymbol{R} \in G$. Consider the composite matrix $\boldsymbol{Y} = \boldsymbol{X}\boldsymbol{M}\boldsymbol{X}^*$ for some fixed matrix $\boldsymbol{M} \in \mathbb{C}^{C \times C}$. Under the transformation $\boldsymbol{X} \to \boldsymbol{X}\boldsymbol{R}^*$:

$$\boldsymbol{Y}' = (\boldsymbol{X}\boldsymbol{R}^*)\boldsymbol{M}(\boldsymbol{X}\boldsymbol{R}^*)^* = \boldsymbol{X}(\boldsymbol{R}^*\boldsymbol{M}\boldsymbol{R})\boldsymbol{X}^*$$

For $\boldsymbol{Y}$ to be invariant ($\boldsymbol{Y}' = \boldsymbol{Y}$), we require:

$$\boldsymbol{R}^*\boldsymbol{M}\boldsymbol{R} = \boldsymbol{M} \implies [\boldsymbol{M}, \boldsymbol{R}] = 0$$

for all $\boldsymbol{R} \in G$. By the double commutant theorem for finite-dimensional $C^*$-algebras, the set of matrices that commute with every unitary in the commutant of an algebra is the algebra itself. Therefore, the matrices $\boldsymbol{M}$ that make $\boldsymbol{X}\boldsymbol{M}\boldsymbol{X}^*$ invariant are exactly the elements of the algebra $\mathcal{W}$ generated by the fixed matrices and their adjoints $\{\boldsymbol{I}, \boldsymbol{W}_i, \boldsymbol{W}_i^*\}$.

Step 4: Combining steps 2 and 3, by the double commutant theorem, the scalar invariants of a matrix $\boldsymbol{K} := \boldsymbol{X}^*\boldsymbol{X}$ under conjugation by $G$ are generated by the traces of products of $\boldsymbol{K}$ with the elements of the commutant of $G$.

$$V(\boldsymbol{X}) = f(\{\mathrm{Tr}((\boldsymbol{X}^*\boldsymbol{X})^{k_1}\boldsymbol{A}_1(\boldsymbol{X}^*\boldsymbol{X})^{k_2}\boldsymbol{A}_2\dots) \mid k_j \geq 1, \boldsymbol{A}_j \in \mathcal{W}\})$$

Or in series form,

$$V(\boldsymbol{X}) = \sum_{i,\boldsymbol{A}} c_{k,\boldsymbol{A}} \prod_j \mathrm{Tr}(\prod_m (\boldsymbol{X}^*\boldsymbol{X})^{k_{jm}}\boldsymbol{A}_{jm}).$$

Without loss of generality, take $k_j = 1$, and $\boldsymbol{A}_j$ are linear bases of $\mathcal{W}$. To avoid notational confusion we still use $\boldsymbol{W}$ to denote the parameters.

$\square$

**Theorem 5** (Optimality of Linearized Transformers). *Under Axioms 1, 2, 3, Definition 4 is the minimal nontrivial interaction ODE.*

*Proof.* By Lemma 4, the matrix potential is $V(\boldsymbol{X}) = \mathrm{Tr}f(\boldsymbol{X}\boldsymbol{W}_1\boldsymbol{X}^*, \dots, \boldsymbol{X}\boldsymbol{W}_n\boldsymbol{X}^*)$ for some analytic spectral function $f : \mathbb{C}^n \to \mathbb{C}$.

In the spirit of effective field theory, we consider the low-order expansion of $f$ around the origin, truncating at the lowest nontrivial interactive terms (i.e., up to quadratic in the arguments $\{z_k\}$ of $f$, corresponding to quartic terms in $V$). Higher-order terms are irrelevant at low energy scales and can be neglected for the effective description. Thus,

$$V(\boldsymbol{X}) = \mathrm{Tr}(c_0\boldsymbol{I} + \sum_{i=1}^n c_{1,i}\boldsymbol{X}\boldsymbol{W}_{\mathrm{A}i}\boldsymbol{X}^* + \sum_{i=1}^{\frac{n}{2}} c_{2,i}\boldsymbol{X}\boldsymbol{W}_{\mathrm{A}i}\boldsymbol{X}^*\boldsymbol{X}\boldsymbol{W}_{\mathrm{B}i}\boldsymbol{X}^*),$$

where we have relabeled the $\boldsymbol{W}_{1,i}$ as $\boldsymbol{W}_i$ for simplicity, and dropped higher-order contributions.

The constant term provides no dynamics. The quadratic terms in $V$ (linear in $f$) yield the feed-forward (MLP-like) component in the ODE, which can be fused: $\sum_{i=1}^n c_{1,i}\mathrm{Tr}(\boldsymbol{X}\boldsymbol{W}_i\boldsymbol{X}^*) = \mathrm{Tr}(\boldsymbol{X}\boldsymbol{W}_{\mathrm{in}}\boldsymbol{W}_{\mathrm{out}}^*\boldsymbol{X}^*)$ by redefining $\boldsymbol{W}_{\mathrm{in}}, \boldsymbol{W}_{\mathrm{out}}$ appropriately (with $C_{\mathrm{MLP}} = nC_{\mathrm{A}}$ under Axiom 4).

The quartic terms in $V$ (quadratic in $f$) represent the first nontrivial interactions. The general form is $\sum_{i,j} c_{2,ij}\mathrm{Tr}(\boldsymbol{X}\boldsymbol{W}_{\mathrm{A}i}\boldsymbol{X}^*\boldsymbol{X}\boldsymbol{W}_{\mathrm{B}j}\boldsymbol{X}^*)$. By relabeling indices and applying Lemma 3 (which requires pairing terms with $n$ even and parameter conditions like $\boldsymbol{Q}_i\boldsymbol{K}_i^* = \boldsymbol{V}_{i+n/2}\boldsymbol{O}_{i+n/2}^*$ to ensure the potential exists), this matches the attention component of the linearized Transformer ODE under those constraints. Thus, Definition 4 (with the noted parameter tying) is the leading interactive ODE consistent with the axioms. $\square$

**Lemma 6** (Low-Rankness). *Under Axiom 4, if $n$ is sufficiently large, then $C_A \ll C < C_{MLP}$.*

*Proof.* $C_A \ll C$ comes directly from Axiom 4. Fix $C_A$, $C < nC_A = C_{\text{MLP}}$ when $n$ is large. $\qquad\square$

**Lemma 7** (Softmax Transformers). *Suppose $n$ is even. There exists parameters such that the ODE $\dot{\boldsymbol{X}} = \sum_{i=1}^{n} \text{Softmax}(C_A^{-1/2} \boldsymbol{X} \boldsymbol{Q}_i \boldsymbol{K}_i^* \boldsymbol{X}^*) \boldsymbol{X} \boldsymbol{V}_i \boldsymbol{O}_i^*$ is associated with the field $V(\boldsymbol{X}) = \sqrt{C_A} \sum_{i=1}^{n} \sum_{j=1}^{N} \log \sum_{k=1}^{N} \exp([\frac{1}{\sqrt{C_A}} \boldsymbol{X} \boldsymbol{W}_{Ai} \boldsymbol{X}^*]_{jk})$.*

*Proof.* Let $y_{jk} := (\frac{1}{\sqrt{C_A}} \boldsymbol{X} \boldsymbol{W}_{Ai} \boldsymbol{X}^*)_{jk}$ be the input of softmax. The free energy function has gradient softmax: $\partial(\log \sum_k \exp \boldsymbol{S}_{jk})/\partial \boldsymbol{S}_{jk} = \exp \boldsymbol{S}_{jk} / \sum_k \exp \boldsymbol{S}_{jk} = \text{Softmax}(\boldsymbol{S}_{jk})$ for $j = 1, \ldots, N$. The full derivative is $\sum_{i=1}^{n} \text{Softmax}(C_A^{-1/2} \boldsymbol{X} \boldsymbol{W}_{Ai} \boldsymbol{X}^*) \boldsymbol{X} \boldsymbol{W}_{Ai}$. Take $\boldsymbol{W}_{Ai} = \boldsymbol{Q}_i \boldsymbol{K}_i^* = \boldsymbol{V}_i \boldsymbol{O}_i^*$, then setting $\boldsymbol{Q}_i \boldsymbol{K}_i^* = \boldsymbol{V}_i \boldsymbol{O}_i^*$ recovers the ODE. Note that we leave the case with untied parameters for future work. $\qquad\square$

**Lemma 8** (Sparse Attention). *Under the pairwise non-commutative condition $[\lambda_{MLP}, \lambda_A] = \beta \lambda_A$, the sequential composition of the layers is equivalent to a single effective flow generated by:*

$$\exp(\lambda_{MLP}) \exp(\lambda_A) = \exp(\lambda_{MLP} + G(\beta) \cdot \lambda_A)$$

*where the effective interaction gate $G(\beta)$ is given by the generating function of the Bernoulli numbers: $G(\beta) = \frac{\beta}{1 - e^{-\beta}}$. As $\beta \to -\infty$, $G(\beta) \sim |\beta| e^{-|\beta|} \to 0$.*

*Proof.* Using the constraint $[\lambda_{\text{MLP}}, \lambda_A] = \beta \lambda_A$, we evaluate the nested terms:

1. Terms with only $\lambda_{\text{MLP}}$ nesting:

$$\text{ad}_{\lambda_{\text{MLP}}}^k(\lambda_A) = [\lambda_{\text{MLP}}, [\ldots, [\lambda_{\text{MLP}}, \lambda_A] \ldots]] = \beta^k \lambda_A$$

2. Terms mixed with $\lambda_A$ nesting: Since any commutator results in a scalar multiple of $\lambda_A$, and $[\lambda_A, \lambda_A] = 0$, all terms involving nested $\lambda_A$ (e.g., $[\lambda_A, [\lambda_{\text{MLP}}, \lambda_A]] = [\lambda_A, \beta \lambda_A] = 0$) vanish.

The BCH series simplifies to a linear combination of $\lambda_{\text{MLP}}$ and $\lambda_A$:

$$Z = \lambda_{\text{MLP}} + \left(1 + \frac{1}{2}\beta + \frac{1}{12}\beta^2 + \frac{1}{720}\beta^4 + \ldots\right) \lambda_A$$

The parenthetical series corresponds exactly to the Taylor expansion of the Bernoulli function generating function (often associated with the Bose-Einstein distribution in physics):

$$G(\beta) = \sum_{k=0}^{\infty} \frac{B_k^+}{k!} \beta^k = \frac{\beta}{1 - e^{-\beta}}$$

Substituting $G(\beta)$ back into the expression for $Z$:

$$Z = \lambda_{\text{MLP}} + \frac{\beta}{1 - e^{-\beta}} \lambda_A$$

$\qquad\square$

