# OpenReview forum: "Transformers are Optimal Effective Fields"
_ICLR.cc/2026/Workshop/Sci4DL — Submitted to Sci4DL 2026_

### Official Review · Reviewer_7T3o · 2026-02-18

**Fit:** 1
**Significance:** 1
**Confidence:** 2

**Summary:**

The paper proposes a theory to explain the design of the Transformer architecture.
By viewing linear Transformers as ODEs, the authors show such ODEs satisfy their proposed optimality axioms.
The authors further discuss the extensions to variants of Transformers and other architectures such as gated MLPs.

**Strengths:**

The raised question on the representational strength of Transformers is clearly relevant to the community.

**Suggestions:**

1. The Sci4DL workshop's primary goal is to use scientific method for empirical analysis of deep learning. This paper however focuses on pure theoretical analysis of Transformers (and other architectural variants). Also the theoretical framework is inspired from variational principles and quantum mechanics, it offers no empirical analysis, not aligned with the call with the workshop.
2. The exposition can be significantly improved:
  - The introduction is confusing: do the authors want to emphasize the perspectives from geometric structures and principle, or the computation power? what existing gaps/open questions this theoretical work sets out to address?
  - The ODE formulation in Defn.1 requires more explanation: it seems that the authors leverages the layer-wise residual connection $X_{t+1} - X_t = \text{GatedMLP}(X_t) + \text{SoftmaxAttn}(X_t)$ to derive the ODE, but it is never explicit. Such formulation was also studied extensively in prior works (e.g. [1], [2]), but the authors fail to acknowledge those.
  - The Related Work section is very poorly written, without proper logical flows.
3. The authors show that linear Transformers already satisfy their proposed optimality axioms (Thm.5). If so, why the use of Softmax attention (which still dominates in practices, and provable theoretical advantages such as expressivity). The theoretical framework fails to distinguish the power of Softmax attention from linear attention, which is a major weakness.

References:
1. Chen, Ricky TQ, et al. "Neural ordinary differential equations." Advances in neural information processing systems 31 (2018).
2. Li, Bei, et al. "ODE transformer: An ordinary differential equation-inspired model for sequence generation." Proceedings of the 60th Annual Meeting of the Association for Computational Linguistics (Volume 1: Long Papers). 2022.

---

### Official Review · Reviewer_QaMh · 2026-02-20

**Fit:** 2
**Significance:** 2
**Confidence:** 2

**Summary:**

The paper proposes an axiomatic and variational framework to derive Transformer-like architectures from first principles. It models a Transformer block as a continuous-time dynamical system over token features and introduces a potential-field formulation in which the dynamics arise as gradients of an invariant potential function. By imposing symmetry and analyticity assumptions on this potential, the authors show that the lowest-order nontrivial interaction terms yield a linearized Transformer with an attention-like structure. The paper further interprets common architectural components—including ReLU-type activations, gated MLPs, softmax attention, and sparse attention—as extensions of the same variational framework, arising through projections, free-energy formulations, or non-commutative corrections. Overall, this work presents a unified perspective in which Transformer design choices emerge from symmetry constraints and low-order interaction expansions.

**Strengths:**

The paper’s main strength lies in its clear and ambitious attempt to provide a principled perspective on Transformer architectures through symmetry and variational reasoning. The axiomatic formulation is conceptually clean and offers a coherent narrative that connects multiple architectural components—attention, activations, and gating—within a single mathematical framework. The use of invariant potentials and low-order interaction expansions provides an intuitive lens that may help readers think about architectural design in a more systematic way, rather than as a collection of empirical heuristics.

**Suggestions:**

The connection between the theoretical framework and practical Transformer architectures is currently limited. The main results are derived in a simplified linearized setting with parameter tying and without several standard components used in modern Transformers. I suggest including a clearer discussion of how the framework applies to realistic architectures, which aspects carry over in practice, and what limitations remain.

The paper also lacks empirical or concrete validation demonstrating the usefulness of the proposed perspective for understanding or designing real models. Including controlled experiments, case studies, or practical examples showing how the framework provides actionable insight would significantly strengthen its impact.

Overall, given that the workshop emphasizes a scientific-method approach—formulating hypotheses and testing them through controlled experiments—the paper could be further improved by incorporating experiments that complement and validate the theoretical framework it proposes.

---

### Meta-Review · Area_Chair_rVvi · 2026-02-28

**Recommendation:** Reject

**Metareview:**

This appears to be a legitimate theoretical paper, but it is not on the *science* of deep learning.

---

### Decision · Program_Chairs · 2026-03-02

Reject